# National spatiotemporal patterns of influenza-like illness in Iran: A pre-COVID-19 study (2015–2019)

**Atieh Sedghian**[1,2]*, **Shahab MohammadEbrahimi**[1,2], **Benn Sartorius**[3,4,5], **Behzad Kiani**[5]

**1** Department of Medical Informatics, School of Medicine, Mashhad University of Medical Sciences, Mashhad, Iran, **2** Student Research Committee, Mashhad University of Medical Sciences, Mashhad, Iran, **3** Centre for Tropical Medicine and Global Health, Faculty of Medicine, University of Oxford, Oxford, United Kingdom, **4** Department of Health Metric Sciences, School of Medicine, University of Washington, Seattle, Washington, United States of America, **5** University of Queensland Centre for Clinical Research (UQCCR), Faculty of Health, Medicine, and Behavioural Sciences, The University of Queensland, Brisbane, Queensland, Australia

* atieh.sedghian1@gmail.com

## Abstract

Lower respiratory tract infections, including Influenza-Like Illness (ILI), contribute significantly to local and global mortality rates. This study aimed to identify high-risk areas for ILI incidence at the county level in Iran during the 4-year period prior to COVID-19. Data were analyzed from 109,167 ILI notifications collected between March 2015 and March 2019 through "The Iran Influenza Surveillance System (IISS)". Spatial scan statistics was utilized to identify significant spatial, temporal, and spatiotemporal clusters of ILI cases. The average age of patients was 40 years (range: 1–117), with females comprising 53% of cases. Males exhibited a higher significant case fatality rate (CFR= 3.28%) compared to females (CFR=2.46%)(P-value<0.05). Among all patients, 53% were tested for influenza, and 22% of those tested had a confirmed case, which accounts for 12% of all ILI patients, with Type A being the predominant strain, accounting for 79.15% of cases. Type A influenza had a higher CFR than type B (5.4% vs. 3.01%, respectively). Among the A virus subtypes, H1N1 exhibited the highest CFR at 8.06%. During the study period, from December 2015 to February 2016, provinces such as Khuzestan, Fars, Kerman, and Yazd reported the highest incidence of ILI. Similarly, the provinces of Alborz, Tehran, and Gilan also experienced elevated ILI rates. In contrast, the period from April 2015 to October 2015 saw the lowest incidence of ILI. Notably, the highest CFR was recorded during the months with the peak ILI incidence. The incidence of ILI fluctuated significantly, peaking at 42.04 in the first year, falling to 24.12 in the second year, and continuing to vary in the following years. These findings underscore the urgent need for tailored public health interventions, such as enhanced surveillance and targeted vaccination campaigns, in provinces where ILI incidence and mortality are highest. By concentrating resources and efforts in these high-risk areas, it may be possible to more effectively manage and control ILI outbreaks.

**Data availability statement:** The data is available in the following repository: "Influenza Like Illness in Iran (2015-2019)", https://doi.org/10.7910/DVN/OGK09I, Harvard Dataverse, V2.

**Funding:** The author(s) received no specific funding for this work.

**Competing interests:** The authors have declared that no competing interests exist.

**Abbreviations:** ILI, Influenza-like Illness; IISS, The Iran Influenza Surveillance System; WHO, World Health Organization; GIS, Geographic Information System; LLR, Log-Likelihood Ratio; CFR, Case Fatality Rate

## 1. Introduction

The World Health Organization's (WHO) Global Burden of Disease identifies lower respiratory tract infections as the fourth leading cause of death [1]. Influenza-like illness (ILI) is defined by WHO as an acute respiratory infection with a fever ≥ 38°C and a cough that started within the last ten days [2]. Only 35–45 percent of ILI cases are caused by influenza, and many cases are triggered by viruses such as respiratory syncytial viruses, rhinoviruses, human coronaviruses, parainfluenza viruses and adenoviruses. Additionally, bacteria such as Mycoplasma pneumonia, Legionella, Chlamydia pneumonia, and Streptococcus pneumonia are the less common causes of ILI [3]. Due to the rapid transmission of the disease and frequent changes in virus antigens, ILI is highly contagious and can lead to global concern [2]. Based on the Global Burden of Disease estimation, 3–5 million severe cases and around 290,000–650,000 respiratory deaths occur annually [4]. Considering the previous literature, ILI has variant spatiotemporal patterns across the different areas, and spatial trends can fluctuate across time frames [1,5].

Essential information on data distributions and patterns in space and time is obtained through spatiotemporal pattern analysis [6]. Because of the similarity of the characteristics of neighboring geographical areas, taking spatial and temporal correlations into account is expected to provide a more accurate estimate of disease-related indicators. It is also possible to determine areas with a high risk of disease (clusters) [7,8]. Exploring the spatiotemporal aspects of the disease can help decision-makers and researchers better understand the disease's pattern and is also critical in executing disease control programs [9].

Research unpacking the spatiotemporal epidemiology of ILI is rich and growing in the world, suggesting strong heterogeneity across and within countries. For instance, a study was conducted in the United States (2021) to identify the hidden patterns of ILI epidemics, and the results showed that season and geographic region affect the spread of ILI, which reaches its peak in winter and spring [10]. Another study conducted in 2019 reported that low temperature and relative humidity are associated with increased virus transmission [11]. Zhang et al. [12] demonstrated people living near public transportation systems are more vulnerable to respiratory infectious diseases. On the other hand, birds and other animals can transmit the virus to rivers or seas, which makes people living around these waters more at risk. In another study, real-time spatiotemporal analysis of micro epidemics of Influenza and COVID-19 [13] was done to recognize patterns of epidemic spread at the community level. They found that if the influenza outbreak was high in one area, it would be 1.9 times more likely to become a center of COVID-19 in the future. As a result, there was a correlation between the influenza and COVID-19 [13].

There are a few studies conducted in Iran related to spatial epidemiology of influenza. A retrospective study (2020) analyzed the spatial distribution of ILI patients at the provincial level and identified six provinces as high incidence of influenza [9]. Another study in Iran (2022), focusing solely on spatial analyses, revealed that living in areas with high population density is connected to the quick transmission of influenza infections. The research also found that certain factors, including comorbidities, age, and gender, influenced the infection [14]. Since ILI is an acute disease with widespread morbidity and mortality and significant economic consequences, proper planning is necessary to reduce the severe effects of ILI epidemics [15,16]. By studying the spatiotemporal distribution of ILI clusters at the county level, important information can be obtained regarding its spread, which could be used to identify high-risk areas and focus on preventive measures. However, no studies have examined the spatial and temporal patterns of ILI incidence in Iran at the county level, while one provincial-level study in Iran suggested that smaller units should be investigated for more accurate results [9]. This study aims

to investigate spatiotemporal patterns of ILI across Iran's counties from March 2015 to March 2019.

## 2. Methods

The Mashhad University of Medical Sciences ethical committee approved this study with the reference number of IR.MUMS.MEDICAL.REC.1401.032. The authors retrospectively accessed the data on 15 January 2023. All data were fully anonymized, ensuring that individuals could not be identified. As a result, the ethical committee waived the requirement for informed consent.

### 2.1 Study area

Iran (S1 Fig), located in the Middle East within Western Asia, exhibits significant climate variability attributed to several influential factors, such as its vast geographical span across various latitudes and the presence of natural features like the Elburz Chain in the north and the Zagros Chain in the west [17]. These varying climate and temperature conditions across different regions of Iran can potentially influence the incidence, seasonality, and transmission of ILI.

Furthermore, it is important to note that Iran experiences social and economic inequalities across its cities, counties, and provinces [18]. These disparities can have implications for health outcomes and the vulnerability of individuals to diseases like ILI. Variations in income, access to healthcare, education, and infrastructure may contribute to different levels of susceptibility and impact from ILI within various communities.

As of 2023, its population exceeds 85 million people, residing in 31 provinces, 398 counties, and 1,203 cities, making Iran the 18th most populous country globally [19,20]. The average population density in Iran was recorded at 49.1 individuals per square kilometer in 2016 [21].

### 2.2 Data sources

Two different data sources were used in the current study (S1 Table). First, the ILI data were obtained retrospectively from the "Iranian Influenza Surveillance System (IISS) "from 21 March 2015–22 March 2019. During this time, 109,919 ILI notifications were registered by the IISS. However, 752 cases were excluded due to a missing residency address. Therefore, the spatiotemporal analysis was performed using data from 109,167 ILI notifications at the County level. It is important to note that the same patient could be clinically diagnosed multiple times, as the data represent the number of ILI notifications, not necessarily the number of unique patients. The diagnostic test was conducted solely to differentiate between influenza and non-influenza cases and was performed only for patients suspected of having influenza, a notifiable disease. Real-time reverse transcription-polymerase chain reaction (RT-PCR) assays were used exclusively for influenza detection, with positive cases further classified by pathogen type (A-type [H1N1, H3N2] or B-type). As a result, the diagnoses in this study have been classified into two categories: influenza and other respiratory infections. Second, census population data by country for 2016 were obtained through Iran's Population and Housing Census [19].

### 2.3 Spatiotemporal analysis

We utilized spatial scan statistics to examine the distribution of ILI cases by county and month using SaTScan. Geographic coordinates (latitude and longitude) for each county's centroid were utilized, and population data from the 2016 census were incorporated to account for population effects in identifying high-risk clusters. To assess the significance of clusters, we

considered a significance level of < 0.05. In this study, we applied a Poisson model to analyze the distribution of patients.

**2.3.1 Purely spatial scan statistic.** In our study, we set the maximum cluster size at 50% of the population, determining the maximum proportion of the population at risk within the scan window. Consequently, numerous overlapping windows of varying sizes were generated, collectively covering the entire study area, with each circular window representing a potential cluster.

**2.3.2 Purely temporal scan statistic.** We calculated the number of notifications for each county monthly. The analysis was conducted separately for each year and collectively for the entire four-year period. Using SaTScan software, we applied temporal scanning statistics, defining the temporal window flexibly to detect clusters of varying durations. By identifying these temporal clusters, we were able to pinpoint specific time intervals where the incidence of ILI was unusually high, providing insights into potential outbreaks or seasonal trends.

**2.3.3 Space-time scan statistic.** We employed space-time statistics to detect clusters in both spatial and temporal dimensions. This methodology utilizes space-time scanning statistics, which involve a cylindrical window characterized by a circular, elliptical, or networked geographic base, with its height representing time [22]. The cylindrical window moves simultaneously in space and time, traversing every possible geographic location and size, as well as all potential time frames. For our analysis, the temporal unit was defined at the month level, allowing us to capture monthly variations in the data. As a result, an infinite number of superimposed cylinders with varying sizes and shapes covered the study area, with each cylinder representing a potential cluster.

## 3. Results

Between March 2015 and March 2019, there were 109,167 notifications of ILI in Iran. Table 1 lists the total number of ILI cases by gender, age group, and mortality rates during the study period. The Mean age of the patients was 40 years (range: 1–117) and women accounted for 53% of the cases (n = 58,164). Among all patients, 53% were tested for influenza, and 22% of those tested had a confirmed case, which accounts for 12% of all ILI patients (n=12,784). Of the type of influenza virus (A and B), type A was the most contagious, accounting for 79.15% cases compared to 20.24% for type B. Additionally, type A had a higher case fatality rate (CFR) than type B (5.4% vs. 3.01%, respectively). With the A virus subtypes, H1N1 had the highest CFR at 8.06%, while H3N2 was responsible for the highest percentage of infections (n=4,825, 47.69%).

The CFR increases with age, with the lowest rate observed in children under 5 years (2.06%) and the highest rate in individuals over 75 years (5.07%). This trend underscores the increasing severity and risk of fatality associated with older age groups. The highest incidence rate is observed in the age group 6–49 years (58 per 100,000), followed by children under 5 years (24.35 per 100,000). The incidence rate decreases with age but slightly increases again in individuals over 75 years (18.08 per 100,000). Males had a higher CFR of 3.28% (1,696 deaths/ 51,755 notifications) while females had a CFR of 2.46% (1,432 deaths/ 58,164 notifications). This difference was statistically significant (P-value < 0.05).

### 3.1 Purely temporal analysis

Table 2 presents the results of the year-by-year analysis of all ILI cases reported in Iran during the study period. For example, the analysis shows that the highest incidence of ILI in 2015 occurred during time frames 9, 10, and 11 (December 2015 to February 2016), while the lowest incidence occurred during time frames 1–6 (April 2015 to October 2015). This analysis reveals distinct seasonal patterns in incidence on a year-by-year basis.

**Table 1. The ILI incidence and mortality information referring to Iranian Influenza Surveillance System (IISS), March 2015-March 2019.**

| | Male (%) | Female (%) | Death (%) | Recovery (%) | Total (%) |
|---|---|---|---|---|---|
| Cases: | 51,755 (47.08) | 58,164 (52.91) | 3,128 (2.84) | 106,791 (97.15) | 109,919 (100) |
| Age range: | | | | | |
| ≤5 | 11,493 (22.21) | 7,541 (12.97) | 392 (12.53) | 18,642 (17.46) | 19,034 (17.32) |
| 6-49 | 20,081 (38.80) | 25,254 (43.42) | 973 (31.11) | 44,362 (41.54) | 45,335 (41.24) |
| 50-64 | 8,214 (15.87) | 10,634 (18.28) | 592 (18.92) | 18,256 (17.09) | 18,848 (17.15) |
| 65-74 | 4,820 (9.31) | 6,474 (11.13) | 411 (13.14) | 10,883 (10.19) | 11,294 (10.28) |
| ≥75 | 7,117 (13.75) | 7,012 (12.05) | 716 (22.89) | 13,413 (12.56) | 14,129 (12.85) |
| Not reported | 30 (0.06) | 1,249 (2.15) | 44 (1.41) | 1,235 (1.16) | 1,279 (1.16) |
| Deceased unit: | | | | | |
| CCU | 90 (0.17) | 75 (0.13) | 165 (5.27) | 0 (0.00) | 165 (0.15) |
| ICU | 1,034 (2.00) | 894 (1.54) | 1,928 (61.64) | 0 (0.00) | 1,928 (1.75) |
| Other Units | 572 (1.11) | 463 (0.80) | 1,035 (33.09) | 0 (0.00) | 1,035 (0.94) |
| Virus type: | | | | | |
| A | 4,601 (79.77) | 5,517 (78.63) | 554 (86.56) | 9,564 (78.75) | 10,118 (79.15) |
| H1N1 | 1,792 (38.95) | 2,041 (36.99) | 309 (55.78) | 3,524 (36.84) | 3,833 (37.88) |
| H3N2 | 2,138 (46.47) | 2,687 (48.70) | 161 (29.06) | 4,664 (48.77) | 4,825 (47.69) |
| H9 | 0 (0.00) | 1 (0.02) | 0 (0.00) | 1 (0.01) | 1 (0.01) |
| Other subtyped | 2 (0.04) | 3 (0.05) | 0 (0.00) | 5 (0.05) | 5 (0.05) |
| Un-subtyped | 255 (5.54) | 305 (5.53) | 33 (5.96) | 527 (5.51) | 560 (5.53) |
| Not reported | 414 (9) | 480 (8.71) | 51 (9.20) | 843 (8.82) | 894 (8.84) |
| B | 1,126 (19.52) | 1,462 (20.83) | 78 (12.19) | 2,510 (20.67) | 2,588 (20.24) |
| Not reported | 41 (0.71) | 37 (0.53) | 8 (1.25) | 70 (0.58) | 78 (0.61) |
| Symptoms:* | | | | | |
| fever ≥ 38°C | 39,848 (77.00) | 43,949 (75.56) | 2,184 (69.82) | 81,613 (76.42) | 83,797 (76.23) |
| Sore throat | 14,157 (27.35) | 17,538 (30.15) | 581 (18.57) | 31,114 (29.13) | 31,695 (28.83) |
| Dyspnea | 11,427 (22.08) | 15,933 (27.39) | 391 (12.50) | 26,969 (25.25) | 27,360 (24.89) |
| Hemoptysis | 17,691 (34.18) | 23,248 (39.96) | 689 (22.02) | 40,250 (37.69) | 40,939 (37.24) |
| Chest pain | 7,292 (14.09) | 10,279 (17.67) | 332 (10.61) | 17,239 (16.14) | 17,571 (15.98) |
| Low blood pressure | 13,979 (27.01) | 16,353 (28.11) | 695 (22.21) | 29,637 (27.75) | 30,332 (27.59) |
| Diagnosis: | | | | | |
| Influenza | 5,768 (11.14) | 7,016 (12.06) | 640 (20.46) | 12,144 (11.37) | 12,784 (11.63) |
| Other respiratory infections | 45,987 (88.86) | 51,148 (87.94) | 2,488 (79.54) | 94,647 (88.63) | 97,135 (88.37) |
| Sample: | | | | | |
| Induced phlegm | 121 (0.23) | 154 (0.26) | 5 (0.16) | 270 (0.25) | 275 (0.25) |
| Tracheal tube suction | 376 (0.72) | 351 (0.60) | 175 (5.59) | 552 (0.52) | 727 (0.66) |
| Nasal swab | 5,422 (10.48) | 7,255 (12.47) | 175 (5.59) | 12,502 (11.71) | 12,677 (11.53) |
| Throat swab | 27,326 (52.80) | 29,826 (51.28) | 2,096 (67.01) | 55,056 (51.55) | 57,152 (51.99) |
| Whole lung lavage (WLL) | 17 (0.03) | 19 (0.03) | 7 (0.22) | 29 (0.03) | 36 (0.03) |
| Spinal fluid | 29 (0.06) | 21 (0.04) | 4 (0.13) | 46 (0.04) | 50 (0.05) |
| Necropsy | 17 (0.03) | 25 (0.04) | 24 (0.77) | 18 (0.02) | 42 (0.04) |
| Not reported | 18,447 (35.64) | 20,513 (35.27) | 642 (20.52) | 38,318 (35.88) | 38,960 (35.45) |
| Test result for influenza: | | | | | |
| Negative | 21,413 (41.37) | 23,550 (40.49) | 1,537 (49.14) | 43,426 (40.67) | 44,963 (40.91) |
| Positive [14] | 5,768 (11.15) | 7,016 (12.06) | 640 (20.46) | 12,144 (11.37) | 12,784 (11.63) |
| Not reported | 24,574 (47.48) | 27,598 (47.45) | 951 (30.40) | 51,221 (47.96) | 52,172 (47.46) |
| gender: | | | | | |
| Male | _ | _ | 1,696 (54.22) | 50,059 (46.88) | 51,755 (47.08) |
| Female | _ | _ | 1,432 (45.78) | 56,732 (53.12) | 58,164 (52.92) |

*The reason the total percentages in the symptoms section exceed 100% is that several cases overlap, so all overlapping symptoms were considered together.

**Table 2. Purely Temporal analysis, scanning for clusters with high/ Low rates by window sizes 50%, March 2015-March 2019.**

| Clusters type | Cluster year | Time frame (Month) | Observed cases | Expected cases | OE | Annual cases/ 100,000 | RR | LLR | P-Value |
|---|---|---|---|---|---|---|---|---|---|
| ILI (High) | 2015 | 9–12 | 31,232 | 10,951.33 | 2.85 | 3,648.9 | 38.51 | 28,508.59 | <0.001 |
| ILI (High) | 2016 | 20–24 | 15,754 | 7,855.00 | 2.01 | 1,472.5 | 7.12 | 7,039.15 | <0.001 |
| ILI (High) | 2017 | 33–35 | 22,127 | 7,921.50 | 2.79 | 3,446.9 | 6.94 | 14,023.78 | <0.001 |
| ILI (High) | 2018 | 45–47 | 17,125 | 6,443.75 | 2.66 | 2,667.7 | 5.94 | 9,782.45 | <0.001 |
| ILI (Low) | 2015 | 1–6 | 328 | 16,427.00 | 0.020 | 25.5 | 0.010 | 20,935.26 | <0.001 |
| ILI (Low) | 2016 | 14–19 | 2,183 | 9,426.00 | 0.23 | 170.0 | 0.13 | 6,309.41 | <0.001 |
| ILI (Low) | 2017 | 26–31 | 3,531 | 15,843.00 | 0.22 | 275.0 | 0.13 | 10,888.50 | <0.001 |
| ILI (Low) | 2018 | 37–42 | 3,996 | 12,887.50 | 0.31 | 311.2 | 0.18 | 6,748.02 | <0.001 |

OE: Observed/ expected, RR: Relative risk, LLR: Log likelihood ratio, LLI: Influenza like Illness

In contrast, Fig 1 illustrates the seasonal and annual distribution of ILI based on cumulative incidence, considering all years together. The graph highlights overall trends, showing that ILI activity follows a clear seasonal pattern across the entire 48-month study period. Spring and summer consistently exhibit lower incidences, with cumulative incidence values of 10.01 and 3.75, respectively, when compared across all months. In contrast, ILI cases increase in autumn, peaking in winter, with cumulative incidence values of 43.83 and 81.55, respectively. This pattern remains consistent across the study period, yet variations in annual incidence are evident. Notably, the data shows a significantly lower cumulative ILI incidence in 2016 compared to other years, as indicated by the lower curve for 2016. According to S2 Table, the analysis indicates that the highest incidence of ILI occurred during time frames 9, 10, and 11 (December 2015 to February 2016), while the lowest incidence occurred during time frames 1–8 (April 2015 to October 2015).

The highest ILI case fatality rate (CFR) is also associated with the months of highest ILI incidence (S2 Fig).

## 3.2 Purely spatial analysis

Fig 2 presents the overall spatial clusters map of ILI in Iran over a 4-year period (March 2015-March 2019). This map offers valuable insights into the spatial patterns of ILI across Iran. As can be seen, ILI exhibits a higher spatial concentration in northern, central, and southern Iran. Gilan, Fars, Isfahan, Yazd, and Kerman provinces are at a higher risk of ILI. Conversely, the western, northwestern, eastern, and northeastern regions of Iran are at a lower risk of ILI and are identified as cold spots.

## 3.3 Spatiotemporal analysis separated by year

Fig 3 offers valuable insights into the spatial and seasonal dynamics of ILI in Iran, with data separated by year, covering the period from March 2015 to March 2019.

### 3.3.1 Seasonal variations.

- Spring and summer: These seasons predominantly exhibit cold spots, indicative of low ILI activity, with notably more cold spots than hot spots. Over the four-year study period, patient numbers were higher during spring compared to summer (7,855 vs. 2,943 cases). Specifically, Kerman and Yazd Provinces exhibited the most significant clusters during spring, with a relative risk (RR) of 4.55 and an observed-to-expected case ratio of 4.13. In contrast, provinces such as Khuzestan, Kurdistan, Kermanshah, Lorestan, West and East Azerbaijan, Hamedan, Ilam, and Zanjan showed the least likely clusters, with an RR of 0.44 and an observed-to-expected case ratio of 0.50 (p-value < 0.05).

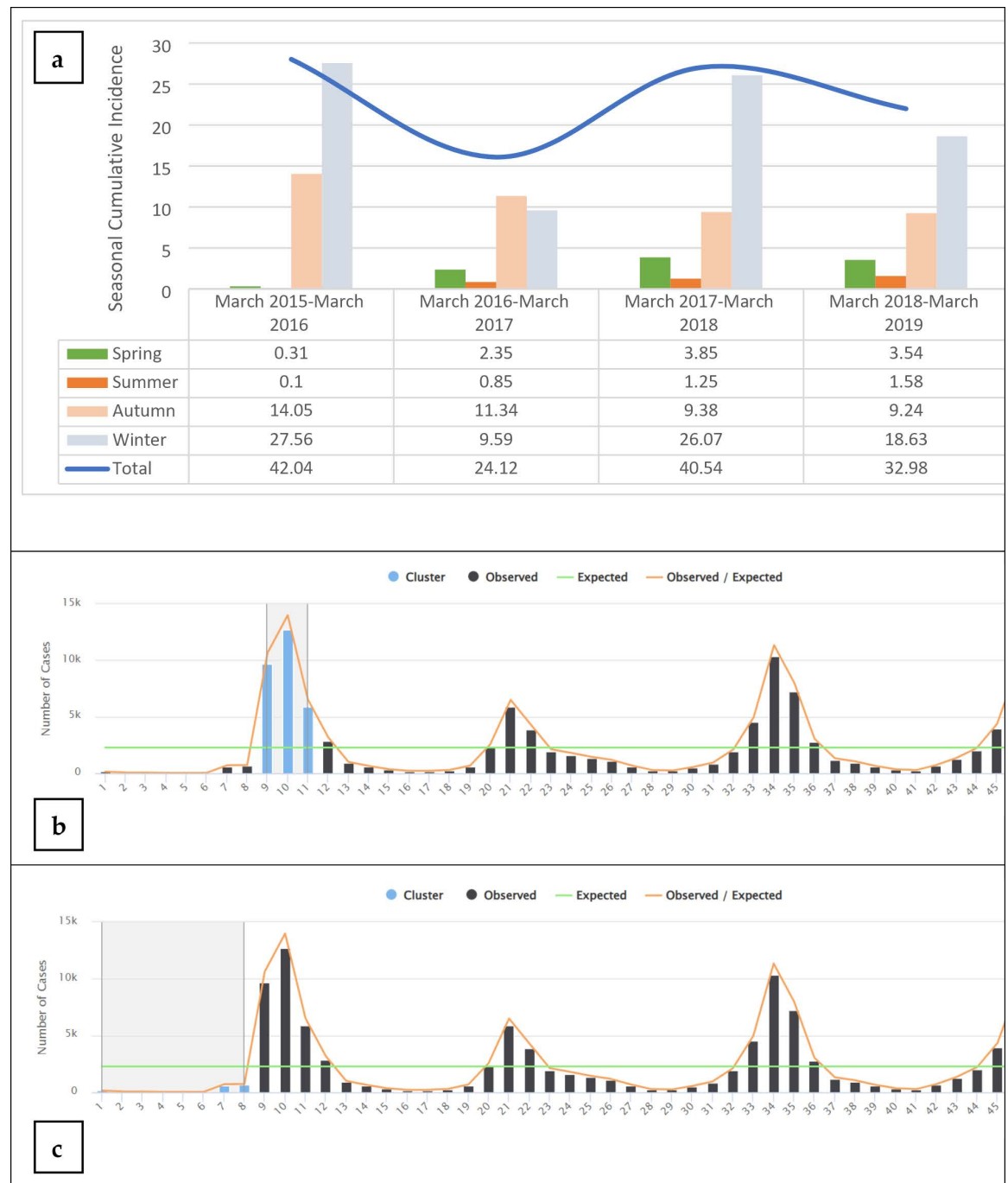

**Fig 1. (a) Trends of seasonal and annual Influenza like Illness cumulative incidence (per 100,000) in Iran from March 2015 to March 2019. (b & c) High and Low temporal clusters over the entire period.**

- Autumn and winter: A significant shift occurs during these seasons, characterized by a pronounced increase in both intensity and concentration of hot spots. Over the four-year study period, patient numbers during winter were 1.8 times higher than in autumn (63,977 vs. 34,392 cases).

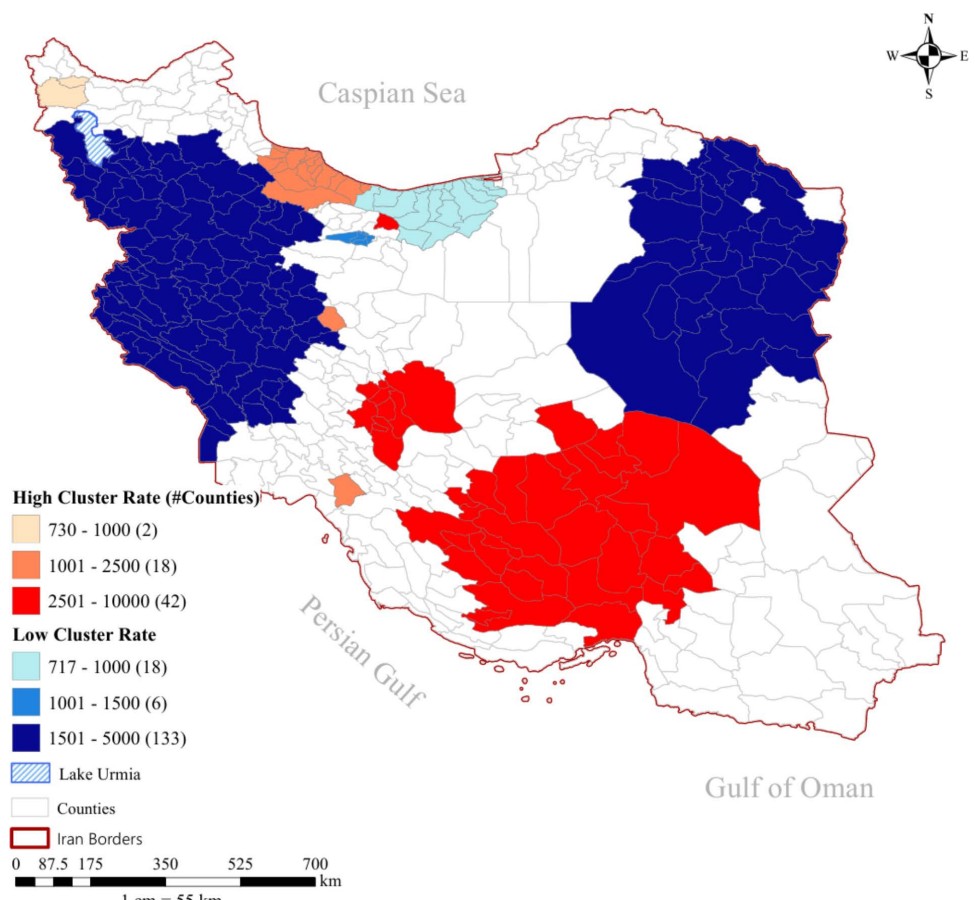

**Fig 2. Most likely and secondary spatial clusters of Influenza like Illness in Iran (March 2015-March 2019) Identification of the clusters is in order of their Log-Likelihood Ratio; Red color clusters indicate the highest likelihood ratio (most likely cluster), while the next clusters (light orange color, sandy color) are secondary from the high-rate clusters.** Vice versa, dark blue color clusters indicate the cluster with the Lowest likelihood ratio (most likely cluster), and others (light and flame blue) are secondary.

During winter over these four years, the most significant clusters were observed in Tehran Province, with an RR of 54 and an observed-to-expected case ratio of 52.32. Kerman and Fars followed with an RR of 3.11 and an observed-to-expected case ratio of 2.78, Isfahan with an RR of 2.36 and an observed-to-expected case ratio of 2.24, and Gilan with an RR of 2.36 and an observed-to-expected case ratio of 2.16. Conversely, the least likely clusters were found in the Khorasan provinces (North and South) with an RR of 0.27 and an observed-to-expected case ratio of 0.28, and in Kermanshah, West and East Azerbaijan, Khuzestan, Hamedan, Markazi, Lorestan, Ilam, and Zanjan with an observed RR of 0.41 and an observed-to-expected case ratio of 0.48 (p-value < 0.05). Winter emerges as the season with the highest concentration of strong hot spots, marking a peak in ILI activity.

**3.3.2 Geographic distribution.** Central, Southern, and Northern Regions: These regions consistently exhibit a higher concentration of ILI throughout the year compared to other areas.

- Cold Spots: Over the study period, most Iranian regions experienced cold spots, with a tendency for these clusters to be concentrated in the western, northwestern, and northeastern

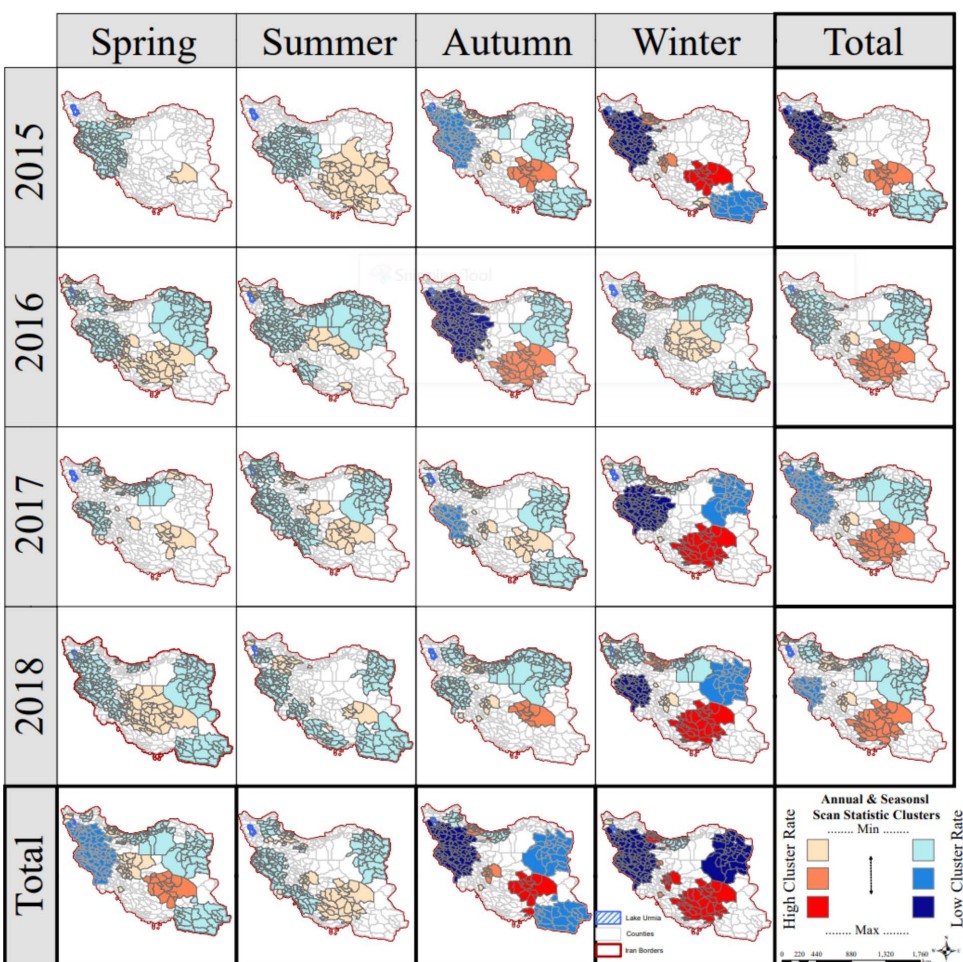

**Fig 3. Most likely and secondary space-time clusters of Influenza like Illness in Iran (March 2015 - March 2019)**
**Identification of the clusters is in order of Log-Likelihood Ratio; Red color clusters indicate the highest likeli-**
**hood ratio (most likely cluster), while the next clusters (light orange color and sandy color) are secondary from**
**the high-rate clusters.** Vice versa, dark blue color clusters indicate the cluster with the Lowest likelihood ratio (most
likely cluster), and others (light and flame blue) are secondary.

parts of the country. This suggests relatively lower ILI activity in these areas compared to
the central and southeastern regions.

- Hot Spots: Conversely, hot spots are predominantly concentrated in the central and south-
  ern regions. Provinces such as Fars, Isfahan, Yazd, and Kerman may bear a heavier burden
  of ILI based on this spatial distribution.

## 3.4 Overall spatiotemporal analysis

The spatiotemporal analysis from March 2015 to March 2019 identified two significant clus-
ters of ILI cases (Fig 4), the first cluster, comprising the provinces of Khuzestan, Fars, Kerman,
Yazd, Isfahan, Bushehr, Hormozgan, Chaharmahal and Bakhtiari, and Kohgiluyeh and Boyer-
Ahmad, exhibited an observed/expected ratio of 6.02 and an RR of 6.47, with highly signifi-
cant p-values. The second cluster, including the provinces of Alborz, Tehran, Zanjan, Qazvin,
Gilan, and Mazandaran, also demonstrated significant results. Both clusters were observed

between December 2015 and February 2016, corresponding to the 9th to 11th months of the study period, as indicated by the temporal analysis. This analysis identified 15 provinces and 151 counties as high-risk areas for ILI.

## 4. Discussion

In this nationwide study, we examined the spatial, temporal, and spatiotemporal patterns among 109,167 ILI notifications in Iran at the county level from March 2015 to March 2019. Populations in a small area are more homogeneous in characteristics than larger scale areas [23]. Therefore, analysis at the county level might be useful in identifying different populations so that interventions can be more specific in these communities. The provinces of Tehran, Kerman, Yazd, Fars, and Isfahan were identified as spatial hotspots. The incidence of ILI was lowest in the summer and highest in the autumn and winter. Overall, spatial hotspot clusters developed more in the central, southern, southeastern, and northern parts during the cold months.

Iran is a geographically diverse nation, spanning from the Caspian Sea in the north to the Persian Gulf in the south, and it experiences four distinct seasons. This variation in climate

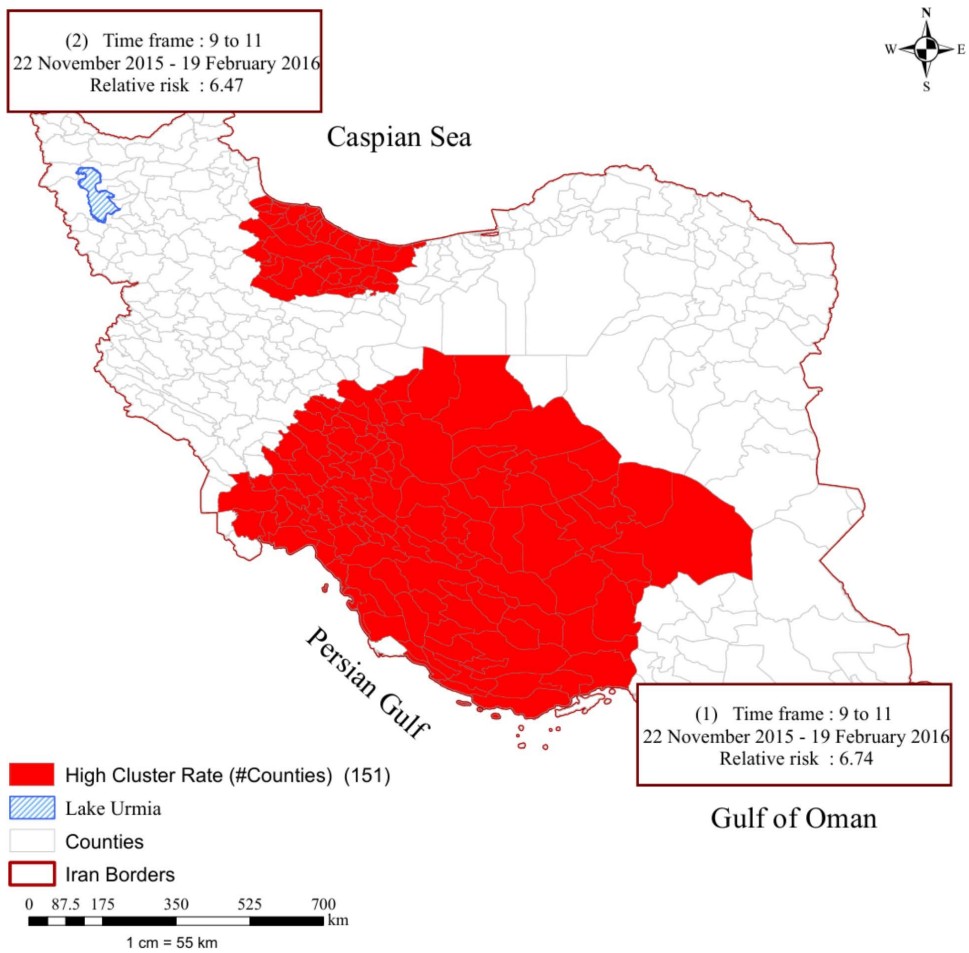

**Fig 4. Most likely space-time clusters of Influenza like Illness in Iran (March 2015-March 2019) identification of the clusters is in order of their Log-Likelihood Ratio; Red color cluster indicates the highest likelihood ratio (most likely cluster).**

results in different levels of relative humidity across different regions [24]. Indeed, this type of climate variation might be related to many kinds of disease occurrences, especially respiratory diseases [25]. Given the intricate U-shaped relationship between relative humidity and ILI incidence [26], and the high averages of rainfall and relative humidity in the coastal areas of the Caspian Sea (north) and the Persian Gulf (south) [24], the detected hotspots in these areas can be justified [27]. Evidence suggests that relative humidity can impact the transmission and severity of ILI [27].

According to meteorological data from the Iranian Statistics Center, the central regions of Iran, including Yazd Province over the four years studied and Isfahan Province in 2015, recorded the lowest levels of precipitation and relative humidity [24]. While dry air by itself cannot result in ILI, it can facilitate the spread and establishment of viruses in the human body. One explanation is that dry air can dry out the mucous membranes in the nose and throat, making them more susceptible to infection [28,29]. Additionally, low humidity levels may make it easier for the ILI viruses to survive and remain airborne [30]. Conversely, some studies have shown that high relative humidity and rainfall levels can also positively affect the transmission of ILI [31–33]. However, further research is needed to fully understand the nature of this relationship and the mechanisms involved.

According to the country's air quality monitoring system, several major industrial cities, including Tehran, Kerman, Isfahan, and Shiraz, which have also been identified as hot spots, have high levels of air pollution [34]. There is some evidence to suggest that exposure to air pollution may increase the risk of respiratory infections, including ILI. Air pollution can irritate the respiratory system, making it more susceptible to infection [35]. Additionally, air pollution can weaken the immune system, making it harder for the body to fight off infections [36,37]. Therefore, in large cities, the spatial heterogeneity of pollutants, along with their inevitable adverse effects on respiratory function, becomes a crucial confounding factor, exacerbating disease outcomes. Consequently, it can predict a poor prognosis for individuals with ILI residing in crowded and polluted areas [38,39]. Given these assumptions, future studies might consider incorporating covariates such as climatic factors and socioeconomic indices into spatiotemporal models.

The study's findings suggest an increased spatial distribution of influenza during colder months (December to February), aligning with prior research on the seasonal behavior of ILI viruses, which typically peak in winter or early spring [40,41]. The emergence of influenza in cold seasons may be attributed to the fact that the coating of ILI viruses becomes more robust at low temperatures, rendering these viruses more active, resistant, and easier to transmit [42]. Moreover, cold and rainy weather can lead to people staying indoors and spending more time in proximity. This situation, particularly in poorly ventilated areas, heightens the risk of transmitting respiratory viruses [43]. Therefore, enhancing indoor air quality (e.g., using humidifiers) and reducing indoor crowding during colder months may play a crucial role in mitigating influenza cases.

The low incidence of ILI in spring and summer can be attributed to high levels of sunlight. There is a relationship between sunlight exposure, vitamin D levels, and the incidence of ILI. Sunlight exposure is a primary sources of vitamin D, an essential nutrient for human health [44]. When the skin is exposed to sunlight, it produces vitamin D, which plays a vital role in the immune system by regulating immune cell function [45]. Evidence suggests that vitamin D can help reduce the risk of ILI by enhancing the production of antimicrobial peptides in the respiratory tract, which can combat viruses and other pathogens before they cause infection [46]. According to meteorological data from the Iranian Statistics Center, the hours of sunshine are significantly higher in spring and summer compared to autumn and winter [24]. This correlation supports the hypothesis that higher sun exposure is associated with a

lower incidence of ILI [47]. To strengthen this relationship, it is recommended to increase public awareness about the benefits of sun exposure, encourage adequate vitamin D intake, and ensure easy access to vitamin D supplements. An additional hypothesis suggests that the low incidence of ILI in summer is due to fewer hospital visits. A study found that during hot seasons, people are less likely to visit hospitals for heat-related illnesses, including respiratory conditions [48].

The observed variation in incidence and CFR of ILI across different age groups underscores the critical need for age-specific public health interventions. These findings highlight the necessity of designing tailored strategies that account for the unique vulnerabilities associated with each age group, which is essential for effectively controlling the spread and impact of ILI. Targeted preventive measures, particularly for high-risk populations, can significantly enhance the efficacy of public health responses, leading to a reduction in both the incidence and mortality rates associated with ILI.

In addition to the aforementioned factors, other key contributors to the transmission of ILI include travel patterns, immunity status, and demographic factors such as age, gender, and underlying health conditions. Regions with a higher rate of tourism or international travel are more susceptible to ILI transmission. The immune status of individuals in various regions can also affect the occurrence of ILI, as factors such as vaccination rates and previous exposure to similar viruses come into play [49–52].

Health policymakers play a crucial role in preventing the spread of ILI by focusing on high-risk areas and populations for vaccination campaigns. To increase public awareness, policymakers can prioritize targeted places such as schools, healthcare facilities, and densely populated urban areas located in high-risk clusters by organizing educational campaigns, public service announcements, and community events to promote vaccination. These recommendations can serve as a framework for future studies aimed at improving ILI prevention and control.

This study boasts several strengths. One of the primary strengths is the large patient sample size, encompassing both inpatients and outpatients, offering reliable and representative data for analysis. Additionally, the spatial and temporal analysis at a partial scale (county level) using the Spatial Scan Statistics method in this study facilitated a more accurate and detailed understanding of the distribution and spread of ILI in Iran.

## 4.1  Limitation

This research has some limitations that should be considered. Firstly, the use of ILI data instead of respiratory disease-specific data means that the study cannot provide specific results and discussions about a particular respiratory disease. Nevertheless, these data offer a broad perspective for researchers working in this area. Secondly, we used the 2016 census data as the reference for population data in our study, which may have affected the generalizability and accuracy of our findings due to its potential out datedness. However, it should be noted that this was the most up-to-date information available for our study, and it was approximately in the middle of our study period, which may have had a minor impact on our results. Furthermore, it is crucial to underscore the substantial impact of the unavailability of 2020 census data due to the Covid-19 pandemic. This circumstance presented a hindrance to our capacity to formulate precise estimations of county populations for each year, drawing on both the 2016 and 2020 census data. Third, the lack of laboratory confirmation for non-influenza respiratory conditions could be regarded as a limitation of this study. Nevertheless, as the primary objective was to analyze the spatiotemporal patterns of ILI at the county level, this limitation is unlikely to have had a significant impact on our aim.

## 5. Conclusions

Our study identifies significant regional variations in the risk of ILI across Iran, with higher risks observed in the central, southern, southeastern, and northern regions. Provinces such as Gilan, Tehran, Fars, Isfahan, Yazd, and Kerman face particularly elevated ILI risks. Temporal analysis shows that high-rate clusters predominantly occur between December and February, while low-rate clusters are more frequent from April to August. Furthermore, spatiotemporal analysis of ILI case fatality rates reveals two significant clusters between December 2015 and February 2016, closely aligning with the timing of peak ILI incidence.

Based on these findings, we recommend implementing targeted interventions in the identified high-risk areas. These interventions should include enhanced public awareness campaigns, increased promotion of personal protective equipment, and expanded vaccination efforts. By considering the observed spatial and temporal patterns, we can optimize responses to ILI and improve public health outcomes.

## Supporting information

**S1 Table. Detailed information about the data sources.**
(DOCX)

**S2 Table. Comprehensive Month-ID Overview for the study period.**
(DOCX)

**S1 Fig. Map of the study area: (a) Iran's location in the Middle East.** (b) Iran's population density per square kilometers (Geometrical Interval)- The colors ranging from pale yellow to dark brown indicate the population density of different counties, based on the 2016 census data. (c) The map shows the 398 counties of Iran (Geometrical Interval), with colors ranging from pale yellow to dark brown indicating the Influenza like Illness incidence rate from March 2015 to March 2019. The identification codes for the study locations (counties) are presented in each polygon. The maps have been created using ArcGIS software.
(TIF)

**S2 Fig. High and Low ILI CFR over the entire period.**
(TIF)

## Acknowledgements

The authors would like to acknowledge the assistance of the Ministry of Health and Medical Education's Center for Communicable Disease Control (CDC) and all the clinicians involved in reporting infectious cases of influenza; without whose support this research would not have been possible.

## Author contributions

**Conceptualization:** Behzad Kiani.

**Data curation:** Atieh Sedghian.

**Formal analysis:** Atieh Sedghian, Shahab MohammadEbrahimi, Behzad Kiani.

**Funding acquisition:** Behzad Kiani.

**Methodology:** Atieh Sedghian, Behzad Kiani.

**Project administration:** Behzad Kiani.

**Software:** Atieh Sedghian.

**Supervision:** Behzad Kiani.

**Visualization:** Atieh Sedghian.

**Writing – original draft:** Atieh Sedghian, Shahab MohammadEbrahimi.

**Writing – review & editing:** Shahab MohammadEbrahimi, Benn Sartorius, Behzad Kiani.

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
