## [Decision Letter · Decision Letter 0]

17 Jan 2025

PONE-D-24-58621National spatiotemporal patterns of influenza-like illness in Iran: A pre-COVID-19 study (2015-2019)PLOS ONE

Dear Dr. Sedghian,

Thank you for submitting your manuscript to PLOS ONE. After careful consideration, we feel that it has merit but does not fully meet PLOS ONE’s publication criteria as it currently stands. Therefore, we invite you to submit a revised version of the manuscript that addresses the points raised during the review process.

A clear presentation of case definition (either clinical or lab-based) and correct interpretation of results based on the type of samples you have used is mandatory for acceptance of th epaper. 

We look forward to receiving your revised manuscript.

Kind regards,

Sana Eybpoosh

Academic Editor

PLOS ONE

Journal Requirements:

3. We note that Figures 2-4 & S1 in your submission contain [map/satellite] images which may be copyrighted. All PLOS content is published under the Creative Commons Attribution License (CC BY 4.0), which means that the manuscript, images, and Supporting Information files will be freely available online, and any third party is permitted to access, download, copy, distribute, and use these materials in any way, even commercially, with proper attribution. For these reasons, we cannot publish previously copyrighted maps or satellite images created using proprietary data, such as Google software (Google Maps, Street View, and Earth). For more information, see our copyright guidelines: http://journals.plos.org/plosone/s/licenses-and-copyright.

a. You may seek permission from the original copyright holder of Figures 2-4 & S1 to publish the content specifically under the CC BY 4.0 license. 

Reviewers' comments:

Reviewer's Responses to Questions

**Comments to the Author**

1. Is the manuscript technically sound, and do the data support the conclusions?

Reviewer #1: Yes

Reviewer #2: Partly

2. Has the statistical analysis been performed appropriately and rigorously? 

Reviewer #1: Yes

Reviewer #2: Yes

3. Have the authors made all data underlying the findings in their manuscript fully available?

Reviewer #1: Yes

Reviewer #2: Yes

4. Is the manuscript presented in an intelligible fashion and written in standard English?

Reviewer #1: Yes

Reviewer #2: Yes

5. Review Comments to the Author

Reviewer #1: The authors reported the spatiotemporal patterns of influenza-like illness in Iran. They appropriately discussed the regional differences in the epidemic situation and the reasons for these differences. I found this to be a highly significant study.

I have two questions for the authors.

In Table 1, the test results are described, and about half of the results were negative. What kind of tests were used for these results? For example, even for rapid influenza virus tests, I think the sensitivity is too low.

Additionally, regarding the types of influenza viruses, nearly 90% are categorized as "other" rather than type A or B. Please provide a breakdown of the "other" category.

Reviewer #2: The manuscript "National Spatiotemporal Patterns of Influenza-Like Illness in Iran: A Pre-COVID-19 Study (2015–2019)" explores the geographic and temporal trends of influenza-like illness in Iran, providing valuable insights into pre-pandemic disease dynamics and public health preparedness.

The manuscript is well written; however, I have the following concerns.

(a) Under the sub-title methods, the authors stated in Lines 114 and 115 that Diagnoses in this study were determined through differential diagnosis based on clinical symptoms and assessments. No laboratory tests were performed to confirm these diagnoses.

1. Clarification of Methods needed

• The authors should provide explicit details on how the listed microorganisms or infections were diagnosed without laboratory testing.

• If laboratory assays were conducted, the methods should be amended to reflect this, including the protocols used.

2. Revision of Results Section:

• The authors should justify the inclusion of specific infections and sample types in the results, correlating these with the stated methodology.

• If certain infections were inferred based on clinical criteria, the diagnostic criteria for each should be detailed.

3. Limitations:

• The limitations of relying solely on clinical diagnoses should be transparently discussed. The authors should highlight the potential for diagnostic inaccuracies and acknowledge the absence of laboratory confirmation.

4. Consider Alternative Approaches:

• If laboratory assays were not done, the authors should limit their claims to broad categories of diseases (e.g., "respiratory infections") rather than naming specific pathogens.

(b)The influence of environmental and socio-economic factors is mentioned but not integrated into the analysis explicitly. The authors should consider incorporating covariates like climate variables (e.g., temperature, and humidity) and socio-economic indicators (e.g., healthcare access) into the spatiotemporal model.

(c)Table I has Men (%) and Women (%) as categories. My concern is that are the children under 5 years also classified as men and women. How can this be married with the gender classification at the end of the table as male and female? These make the table very difficult to comprehend.

6. PLOS authors have the option to publish the peer review history of their article (what does this mean? ). If published, this will include your full peer review and any attached files.

**Do you want your identity to be public for this peer review?** For information about this choice, including consent withdrawal, please see our Privacy Policy .

Reviewer #1: **Yes: ** Yutaro Akiyama

Reviewer #2: **Yes: ** Innocent Afeke

---

## [Author Response · Author response to Decision Letter 0]

5 Feb 2025

Response to Reviewers

Dear editor,

We are grateful for your consideration of this manuscript, and we very much appreciate your suggestions, which have been very helpful in improving the manuscript. We also thank the reviewers for their careful reading of our text and providing invaluable comments.

All the comments we received on this study have been taken into account and addressed or responded. In this letter, we present our reply to each of them separately.

Two versions of the manuscript are enclosed, one where all the changes have been track-changed, and another version without any marks.

We hope that these changes to the manuscript will facilitate the decision to publish this study in your journal. We are open to consideration of any further comment on our answers.

Sincerely,

The authors

Journal Requirements:

Authors’ answer: Thank you for providing your feedback and guidance regarding our manuscript. We have reviewed the journal's style requirements and ensured that the manuscript complies fully with PLOS ONE's formatting and style guidelines. Additionally, all file names have been updated to meet the specified criteria.

Authors’ answer: Thank you for the reminder regarding the requirement for an ORCID for the corresponding author. I confirm that I have completed this step and successfully validated my ORCID in the Editorial Manager system.

3. We note that Figures 2-4 & S1 in your submission contain [map/satellite] images which may be copyrighted. All PLOS content is published under the Creative Commons Attribution License (CC BY 4.0), which means that the manuscript, images, and Supporting Information files will be freely available online, and any third party is permitted to access, download, copy, distribute, and use these materials in any way, even commercially, with proper attribution. For these reasons, we cannot publish previously copyrighted maps or satellite images created using proprietary data, such as Google software (Google Maps, Street View, and Earth).

Authors’ answer: Thank you for bringing this to our attention. To ensure compliance with PLOS ONE's copyright policy, we have removed the satellite layers from the maps in Figures 2–4 and S1. The updated figures now utilize openly available data and sources that align with the Creative Commons Attribution License (CC BY 4.0).

Editor comment:

A clear presentation of case definition (either clinical or lab-based) and correct interpretation of results based on the type of samples you have used is mandatory for acceptance of the paper.

Authors’ answer:

Thank you for this comment. After consulting with the data administrator, we confirmed that the diagnostic test was conducted solely to differentiate between influenza and non-influenza cases. It was not performed for all patients but only for those for whom the doctor suspected of having influenza, a notifiable disease, and deemed testing necessary. Laboratory testing, using real-time reverse transcription-polymerase chain reaction (RT-PCR) assays, was exclusively applied to influenza. For positive cases, the specific pathogen type (A-type [H1N1, H3N2] or B-type) was recorded, and we categorized our results accordingly.

For patients who were not tested or received a negative test result, doctors relied on clinical assessment to make an inferential diagnosis, which we had originally mentioned in the paper. However, since this approach may be subject to bias and is not directly relevant to our primary aim of analyzing the spatiotemporal patterns of influenza-like illnesses (ILI), we decided to categorize these cases as “other respiratory diseases,” in alignment with Reviewer 2’s comment. The maps and most of the paper remain unchanged, with modifications made only to Table 1, specifically in the diagnosis section.

We also made some changes to the methods section to clarify this (Page 5, Lines 106-111):

“The diagnostic test was conducted solely to differentiate between influenza and non-influenza cases and was performed only for patients suspected of having influenza, a notifiable disease. Real-time reverse transcription-polymerase chain reaction (RT-PCR) assays were used exclusively for influenza detection, with positive cases further classified by pathogen type (A-type [H1N1, H3N2] or B-type). As a result, the diagnoses in this study have been classified into two categories: influenza and other respiratory infections.”

Reviewers' comments:

Reviewer #1:

The authors reported the spatiotemporal patterns of influenza-like illness in Iran. They appropriately discussed the regional differences in the epidemic situation and the reasons for these differences. I found this to be a highly significant study.

I have two questions for the authors.

Authors’ answer: Thank you for taking the time to review our manuscript. We have addressed all your comments as follows.

1. In Table 1, the test results are described, and about half of the results were negative. What kind of tests were used for these results? For example, even for rapid influenza virus tests, I think the sensitivity is too low.

Authors’ answer:

We thank you for your insightful comments. As we mentioned in our response to the editor, laboratory testing using RT-PCR was conducted exclusively to confirm influenza cases, and no rapid influenza diagnostic tests were used. Therefore, many patients with influenza-like illness (ILI) tested negative for influenza, as their symptoms were caused by other respiratory infections. The use of RT-PCR to diagnose influenza in Iran was also previously reported in another study [1].

2. Additionally, regarding the types of influenza viruses, nearly 90% are categorized as "other" rather than type A or B. Please provide a breakdown of the "other" category.

Authors’ answer: Thank you for identifying this error in our calculation. Upon reviewing the table, we found that the counts for influenza types A and B included all records rather than being restricted to positive cases. We have now corrected this by including only positive cases in the calculations, resulting in updated values in the relevant table.

Additionally, the category previously labeled as "other" has been revised to "not reported" to more accurately reflect the nature of the data. These revisions have been incorporated into both the manuscript and the updated table. We trust that these adjustments adequately address your concerns (Page 7, Table 1: virus type).

“

Virus type:

A

- H1N1

- H3N2

- H9

- Other subtyped

- Un-subtyped

- Not reported

B

Not reported

4,601 (79.77)

1,792 (38.95)

2,138 (46.47)

0 (0.00)

2 (0.04)

255 (5.54)

414 (9)

1,126 (19.52)

41 (0.71)

5,517 (78.63)

2,041 (36.99)

2,687 (48.70)

1 (0.02)

3 (0.05)

305 (5.53)

480 (8.71)

1,462 (20.83)

37 (0.53)

554 (86.56)

309 (55.78)

161 (29.06)

0 (0.00)

0 (0.00)

33 (5.96)

51 (9.20)

78 (12.19)

8 (1.25)

9,564 (78.75)

3,524 (36.84)

4,664 (48.77)

1 (0.01)

5 (0.05)

527 (5.51)

843 (8.82)

2,510 (20.67)

70 (0.58)

10,118 (79.15)

3,833 (37.88)

4,825 (47.69)

1 (0.01)

5 (0.05)

560 (5.53)

894 (8.84)

2,588 (20.24)

78 (0.61)

“

Reviewer #2:

The manuscript "National Spatiotemporal Patterns of Influenza-Like Illness in Iran: A Pre-COVID-19 Study (2015–2019)" explores the geographic and temporal trends of influenza-like illness in Iran, providing valuable insights into pre-pandemic disease dynamics and public health preparedness.

The manuscript is well written; however, I have the following concerns.

(a) Under the sub-title methods, the authors stated in Lines 114 and 115 that Diagnoses in this study were determined through differential diagnosis based on clinical symptoms and assessments. No laboratory tests were performed to confirm these diagnoses.

1. Clarification of Methods needed:

• The authors should provide explicit details on how the listed microorganisms or infections were diagnosed without laboratory testing. If laboratory assays were conducted, the methods should be amended to reflect this, including the protocols used.

Authors’ answer: We appreciate your thoughtful feedback on the methods section. To address your concerns and provide greater clarity, we would like to offer the following explanation:

As outlined in Table 1, laboratory testing was performed exclusively to confirm influenza cases among respiratory infections. The proportion of notifications with laboratory-confirmed influenza is clearly presented in the table. For all other respiratory conditions, no laboratory tests were conducted. Instead, diagnoses were determined through clinical assessments and differential diagnosis, based on patients’ symptoms and the prevailing epidemiological context.

To ensure complete transparency and resolve any potential ambiguities, we have revised the methods section to explicitly clarify that laboratory testing was limited to influenza diagnosis, while other respiratory conditions were identified solely through clinical evaluation.

We trust this clarification addresses your concerns and enhances the overall clarity and precision of the manuscript (Page 5, Lines 106-111).

“The diagnostic test was conducted solely to differentiate between influenza and non-influenza cases and was performed only for patients suspected of having influenza, a notifiable disease. Real-time reverse transcription-polymerase chain reaction (RT-PCR) assays were used exclusively for influenza detection, with positive cases further classified by pathogen type (A-type [H1N1, H3N2] or B-type). As a result, the diagnoses in this study have been classified into two categories: influenza and other respiratory infections.”

2. Revision of Results Section:

• The authors should justify the inclusion of specific infections and sample types in the results, correlating these with the stated methodology. If certain infections were inferred based on clinical criteria, the diagnostic criteria for each should be detailed.

Authors’ answer: We thank the reviewer for the insightful feedback. The differential diagnosis of respiratory conditions in this study was based on the documented clinical history and symptomatology of patients, without utilizing laboratory assays. Laboratory testing was exclusively employed for the confirmation of influenza cases, and the results of these diagnostic tests, aligned with the sample types detailed in the Results section, are presented in Table 1. To ensure the scientific rigor and credibility of our findings, we have revised the terminology used in the Results section. Specific infections initially mentioned have been consolidated under the broader term "other respiratory infections," as no laboratory assays were performed to confirm these diagnoses. Influenza, being the sole condition for which laboratory confirmation was conducted, remains explicitly stated in both the text and Table 1. These revisions have been incorporated into the manuscript to align the Results section more closely with the study's methodology and the scope of diagnostic testing employed (Page 7, Table 1: Diagnosis)

“

Diagnosis:

Influenza

Other respiratory infections

5,768 (11.14)

45,987 (88.86)

7,016 (12.06)

51,148 (87.94)

640 (20.46)

2,488 (79.54)

12,144 (11.37)

94,647 (88.63)

12,784 (11.63)

97,135 (88.37)

“

3. Limitations:

• The limitations of relying solely on clinical diagnoses should be transparently discussed. The authors should highlight the potential for diagnostic inaccuracies and acknowledge the absence of laboratory confirmation.

Authors’ answer: We thank the reviewer for this valuable comment. The lack of laboratory confirmation for non-influenza respiratory conditions has been acknowledged as a potential limitation of this study. However, given that the primary focus was to examine the spatiotemporal patterns of influenza-like illness across Iran, this limitation is unlikely to have introduced significant bias or affected the overall findings. This point has been explicitly addressed in the revised "Limitations" section to ensure clarity and transparency regarding the scope and implications of our study (Page 16, Lines 328-331).

“Third, the lack of laboratory confirmation for non-influenza respiratory conditions could be regarded as a limitation of this study. Nevertheless, as the primary objective was to analyze the spatiotemporal patterns of ILI at the county level, this limitation is unlikely to have had a significant impact on our aim. “

4. Consider Alternative Approaches:

• If laboratory assays were not done, the authors should limit their claims to broad categories of diseases (e.g., "respiratory infections") rather than naming specific pathogens.

Authors’ answer: We thank the reviewers for insightful comment and fully agree with your observation regarding the specificity of pathogen identification in the manuscript. To address this concern and enhance clarity, we have revised the diagnostic classifications used in the study.

We have updated the diagnostic categories to two broad classifications: influenza and other respiratory infections. This revision ensures that the claims made in the manuscript accurately reflect the study methodology.

These changes have been incorporated into the Results section, with corresponding updates made to the relevant table. This adjustment clarifies that specific pathogen identification, beyond influenza, was not performed, in line with the diagnostic approach employed.

We trust that this revision effectively addresses your concern and enhances the transparency and accuracy of the manuscript (Page 7, Table 1: Diagnosis).

“

Diagnosis:

Influenza

Other respiratory infections

5,768 (11.14)

45,987 (88.86)

7,016 (12.06)

51,148 (87.94)

640 (20.46)

2,488 (79.54)

12,144 (11.37)

94,647 (88.63)

12,784 (11.63)

97,135 (88.37)

“

(Abstract, Lines: 22 to 25)

“Among all patients, 53% were tested for influenza, and 22% of those tested had a confirmed case, which accounts for 12% of all ILI patients, with Type A being the predominant strain, accounting for 79.15% of cases. Type A influenza had a higher CFR than type B (5.4% vs. 3.01%, respectively). Among the A virus subtypes, H1N1 exhibited the highest CFR at 8.06%.”

(Page: 6, Lines: 141-146)

“Among all patients, 53% were tested for influenza, and 22% of those tested had a confirmed case, which accounts for 12% of all ILI patients (n=12,784). Of the type of influenza virus (A and B), type A was the most contagious, accounting for 79.15% cases compared to 20.24% for type B. Additionally, type A had a higher case fatality rate (CFR) than type B (5.4% vs. 3.01%, respectively). With the A virus subtypes, H1N1 had the highest CFR at 8.06%, while H3N2 was responsible for the highest percentage of infections (n=4,825, 47.69%).”

(b) The influence of environmental and socio-economic factors is mentioned but not integrated into the analysis explicitly. The authors should consider incorporating covariates like climate variables (e.g., temperature, and humidity) and socio-economic indicators (e.g., healthcare access) into the spatiotemporal model.

Authors’ answer: We sincerely appreciate your comment regarding the inclusion of environmental and socio-economic factors in our analysis. We would like to provide further clarification on the focus and scope of our study in relation to these factors.

The primary objective of our study was to explore the spatiotemporal patterns of influenza-like illness (ILI) in Iran. Although we have mentioned the potential impacts of environmental factors (such as temperature and humidity) and socio-economic indicators (such as healthcare access) on disease patterns, these variables were not directly i

---

## [Decision Letter · Decision Letter 1]

28 Feb 2025

National spatiotemporal patterns of influenza-like illness in Iran: A pre-COVID-19 study (2015-2019)

PONE-D-24-58621R1

Dear Dr. Atieh Sedghian,

We’re pleased to inform you that your manuscript has been judged scientifically suitable for publication and will be formally accepted for publication once it meets all outstanding technical requirements.

Kind regards,

Sana Eybpoosh

Academic Editor

PLOS ONE

Reviewers' comments:

Reviewer's Responses to Questions

**Comments to the Author**

1. If the authors have adequately addressed your comments raised in a previous round of review and you feel that this manuscript is now acceptable for publication, you may indicate that here to bypass the “Comments to the Author” section, enter your conflict of interest statement in the “Confidential to Editor” section, and submit your "Accept" recommendation.

Reviewer #1: All comments have been addressed

Reviewer #2: All comments have been addressed

2. Is the manuscript technically sound, and do the data support the conclusions?

Reviewer #1: Yes

Reviewer #2: No

3. Has the statistical analysis been performed appropriately and rigorously? 

Reviewer #1: Yes

Reviewer #2: Yes

4. Have the authors made all data underlying the findings in their manuscript fully available?

Reviewer #1: Yes

Reviewer #2: Yes

5. Is the manuscript presented in an intelligible fashion and written in standard English?

Reviewer #1: Yes

Reviewer #2: Yes

6. Review Comments to the Author

Reviewer #1: (No Response)

Reviewer #2: The revised manuscript is significantly improved, with most concerns addressed satisfactorily. The authors have appropriately acknowledged the limitations of relying on ILI data, outdated census information, and the lack of laboratory confirmation. I agree that these limitations do not fundamentally compromise the study’s objectives.

7. PLOS authors have the option to publish the peer review history of their article (what does this mean? ). If published, this will include your full peer review and any attached files.

**Do you want your identity to be public for this peer review?** For information about this choice, including consent withdrawal, please see our Privacy Policy .

Reviewer #1: No

Reviewer #2: **Yes: ** Innocent Afeke

---

## [Editor Report · Acceptance letter]

PONE-D-24-58621R1

PLOS ONE

Dear Dr. Sedghian,

I'm pleased to inform you that your manuscript has been deemed suitable for publication in PLOS ONE. Congratulations! Your manuscript is now being handed over to our production team.

Kind regards,

on behalf of

Dr. Sana Eybpoosh

Academic Editor

PLOS ONE